# Out of Print: What the Pandemic-Era Newspaper Crisis in Australia Teaches Us about the Role of Rural and Regional Newspapers in Creating Sustainable Communities

Olav Muurlink [1,*] and Elizabeth Voneiff Marx [2]

1   School of Business and Law, Central Queensland University, Brisbane 4000, Australia
2   School of Political Science and International Studies, The University of Queensland, Brisbane 4067, Australia
*   Correspondence: o.muurlink@cqu.edu.au

**Abstract:** Print newspapers tend to form part of the conversation on sustainable development goals in terms of the ability to communicate goals to the public, but to what degree are print newspapers part of the solution to sustainable rural and regional communities in particular? The COVID-19 pandemic coincided with a global crisis in print journalism. This article takes Australia as an extreme case study of the collapse of print news, tracing both the immediate causes as well as the scale of the decline, and the impacts in terms of community conversation, building social capital, and improving governance, particularly in sub-populations such as the aged, and in digitally disadvantaged regional and remote communities. This paper uses a range of secondary and primary data sources to build a paradoxical picture of a revival of rural and regional journalism, a revival that is focused on survival rather than revisiting the activist origins of early independent rural and regional media in the country. The new papers include part of the traditional mission of print news—building social capital—but are less engaged in creating political and financial transparency. It is concluded that the new wave of rural and regional titles may be simply at an early stage of evolution, but with the digital divide in Australia reducing, they may have little time to evolve.

**Keywords:** sustainable communities; SDGs; community; print newspapers; entrepreneurship

## 1. Introduction

When scholars discuss sustainable development goals (SDGs) in relation to the media in general, and the presence of print newspapers in particular, the discussion tends to centre around how newspapers frame and communicate those goals (e.g., [1,2]) rather than the direct role newspapers may play in delivering SDG outcomes. How print newspapers might help determine educational, gender equality, health, and even public health outcomes is rarely directly considered, but there is a substantial body of work delineating the relationship between the presence and use of print newspapers in developing community literacy and social capital, and the role that literacy and social capital play in turn, in delivering SDGs. What does print media offer that digital options do not?

It is timely to consider this nexus, as there are a range of clear indicators of a global crisis in print news, particularly in vulnerable rural and remote communities. The capacity of the world's major producers of newsprint has been falling steadily [3], and that fall equates to both a decline in the number of publications, editions, and pages being produced globally. In almost every market, bar India, circulations continued to shrink—ranging from a 15% decline in Japan to a 48% fall in Brazil in the five years from 2013 to 2018 [3]). In fact, the Indian digital news sphere, despite being new to the game, is "already as big as some of Europe's largest markets" thanks to a large and growing educated middle class [3]. Elsewhere, in 2019, in the run-up to the COVID-19 crisis, newspaper circulations continued to fall, with Pew research showing circulation figures in the United States (US), for example, dropping to their lowest level since at least 1940, when reliable records began

to be collated [4]. In some cases, these figures have been more dramatic—particular titles in Australia and internationally lost two-thirds of their audited circulation over a period of just five years [5], leading to restructures, amalgamations, and closures.

Not coincidentally, advertising spending is also falling. In 2019 alone, the global advertising spending on newspapers declined by over US$6 billion, while television, radio, cinema, and outdoor advertising continued to grow, and digital advertising boomed. The share of advertising spending allocated to newspapers has fallen since the year 2000 from over half to a quarter [6]. There is a linkage between ongoing reader shrinkage and cuts to advertising revenue that undercuts the business model of the small to medium-sized newspaper. The newspaper industry is what Angelucci and Cagé call a "canonical example" of a two-sided market [7]), where advertisers and readers form the two 'markets' for the newspaper proprietor, with the characteristics (and presence) of readers determining the characteristics and presence of advertisers, and to a large extent, vice versa. In terms of sustainable communities, active participation by both groups remains imperative.

Collectively, not surprisingly, this rapid downward trend in numbers has translated into an absolute loss of newspaper titles. The loss of iconic titles, including the 168 year old *News of the World* [8] and the Pulitzer-prize winning *Tampa Tribune* and the *Cincinnati Post* [9] are just small examples of closures, mergers, or movement to online-only editions.

If print newspapers are an endangered species in the developed world, then a particularly vulnerable sub-species are regional and remote newspapers where the pool of revenue and readers is smaller. The high-profile closures mentioned previously, however compelling, mask the far greater numbers of small rural or community newspapers which have collapsed and closed in recent years. For small newspapers, operating closer to the margins in terms of advertising sales and readership, declines are lethal. In the US, for example, in the five years to 2018, a net 1779 US-based publications ceased to operate without national fanfare.

Circulation of local, as opposed to state or national, newspapers in the United States declined by 27% from 2003 to 2014 according to Pew research [10], while more recent Australian data show similar patterns [11]. Even more concerningly, between 2013 and 2018, there was a 15% decrease in the number of local and regional newspaper titles in Australia (ACCC analysis). In the ten-year period to 2018 alone, 21 local government areas in that country, the equivalent of counties in the United States or shires in the United Kingdom (UK), were left without even a single local or regional newspaper. In the 2005–2018 period, the Press Gazette reported on the net loss of 245 local news titles in the UK [12], creating what Abernathy refers to as "news deserts" or what might more aptly be called "*newspaper deserts*" [13].

The questions this article poses are: What are the impacts of this decline of rural and regional newspapers (as distinct to other forms of media) on sustainable development goals? Does the disappearance of small, local, and regional newspapers matter, or will social media or large multi-media publishing conglomerates be able to fill the gap? The SDGs address issues of social justice, health, human rights, and economic growth, and in this analysis, we first examine how local print newspapers have a role in securing the aims of sustainable communities. Scholarly and non-scholarly debate on the importance of local print newspapers indicates community sustainability is threatened by the loss of regional titles [14–17]. We first examine what the literature shows happens to a community when it becomes part of the 'newspaper desert' through the loss of local print news and associated newsrooms.

We then examine, as a case study, the unprecedented mass collapse of regional print newspapers that occurred in Australia during 2020, and early signs that small entrepreneurs have moved rapidly to fill the breach, sensitive to the role that community newspapers play in terms of producing sustainable communities. Finally, this analysis shows how, remarkably, the pandemic appears to have revitalized, at least temporarily, independent press in a market that has been traditionally one of the least diverse in the world, and offers caveats on optimism that might arise from this resurgence in relation to the delivery on the SDGs.

## 2. Newspapers and the Sustainability of Communities

The impact of the closure of a newspaper in a discrete rural community is severe and involves the loss of social capital and the loss of the ability to tell and retain the historicity of communities, thus resulting in weakened community identity. The closure of newspapers in regions of low digital literacy, poverty, and poor internet access has consequences that can be understood without a scholarly study to 'prove' impact. Local newspapers are the keystone of local news ecosystems according to Nielsen [18], and when they disappear, the community itself disappears from the radar of central government. (One rural mayor told the authors that one of the reasons a newspaper—whether critical or supportive of the shire council—was important to the shire was that it made the region visible to the state government in Australia. Press clipping services take the outputs of small newspapers and put them on the desks of policy makers and politicians on a state and national level.

The difficulty a community may experience in accessing death notices or statutory advertising relating to government regulations and local developments, the absence of a public clearinghouse of birth and wedding photographs, local sports pictures that can be clipped out and placed in an album of memories without the need for a printer or computer or indeed electricity, and the increased reliance on uncurated news, obscuring the local in a mass of generic, national news and entertainment, means that the community is inevitably less informed about itself.

But is this summary of a priori truisms true?

### 2.1. Social Capital

In *Life is Harder*, the regional editor of a series of daily newspapers describes a recent, extended study of one community in Caroline County, Virginia, which suddenly and without notice, lost a 99 year old newspaper, the *Caroline Progress*, in 2019. Matthews [13] colours in the wounds that accrue to a community when it loses its sole dedicated print newspaper, describing "community members' sense of community, with participants missing celebrated gatherings, suffering from an increased sensation of isolation and diminished pride in the community" (p. v). The sudden shut-down recalls what Sarason influentially wrote of this connection to community: "You know when you have it and when you don't" [19].

Matthews' qualitative study saw him spend extended time in the community mapping the days after the collapse of the paper, noting that older locals desperate for local news turned to purchasing papers sourced from major population centres to fill the gap, accessing or not accessing online news and eventually turning "to what gets printed on the back of their water bill" out of desperation (p. 49). It is interesting to note that Davidson and Cotter [20] conducted a telephone survey of over 1000 county residents in rural United States and found that those rated high on psychological sense of community were more interested in reading the news.

### 2.2. Health

Beyond the perceived loss of community experienced by, in particular, the older members of a community, the literature shows that the shift from print to online and social media has measurable impacts on significant outcomes. International studies show that rural newspapers traditionally played a role in providing accurate access to health information in rural areas [21,22], with the Public Health Association of Australia [23] more broadly expressing concern about the proliferation of inaccurate information facilitated by digital platforms. In print, such messaging is easier to monitor and correct. In a developing-world context, newspaper readership is associated with a higher likelihood to use modern contraceptives, for instance, and increased confidence in healthcare [24].

### 2.3. Governance and Civic Engagement

Local newspapers demonstrably improve governance and are related to markers of the political health of a community. Studies show that local newspapers improve

government accountability and operating efficiency [25,26]. They reduce polarization of voting behaviours by increasing voter access to local news on which to base political decisions [27]. Closures lead to fewer candidates standing for elections and a reduction in voter turnout [9] and lower voter turnout [28]. These impacts are not only observed at the local level. An analysis of both short- and long-term impacts of the closure of the *Cincinnati Post*, the only daily newspaper in its particular market in Ohio, saw fewer candidates stand for the subsequent election and a reduction in voter turnout [8], a finding confirmed elsewhere in the US [29,30]. Even three years after the closure, voter engagement remained impacted. The study found the suppression of civic engagement confirmed in other studies e.g., [31,32], suggesting that digital media does not act as an adequate substitute.

The closure of newspapers certainly leads to a reduction in the quantity of reporting, and competition in the news market, that can have impacts well beyond the regions in terms of ensuring the airing of accurate and diverse content. There is some evidence that the mere presence (rather than the absolute quality) of a newspaper results in improvements; a Danish study showed that even newspapers with low intensity of local government reporting had a positive impact on policy quality [33]. A Tilburg University study longitudinally examined small county newspaper closures in the US and stock price informativeness of firms in the US, finding statistically significant associations between closures and accuracy in corporate information [34], a pathway confirmed separately by Cahan et al. [35].

The net impact of newspaper closures on civic life is clear: the absence of traditional printed newspapers leads to an erosion of local knowledge that digital media fails to address [36], discussed in the next section.

## 2.4. Trustworthy Information and News Diversity

The vacuum left when a newspaper departs may be filled with online media, but digital media is associated with increased distrust in journalism, is often politicised, lacks quality gatekeeping such as subediting, and fails to enhance the physical community (as opposed to online communities). Speakman showed that exposure to online news leads to an increasing distrust in news [37], and Australian regional media consumers trust local print newspapers more than digital (19% to 9%). A Reuters Institute digital news report in 2021 confirmed these attitudes persist [38]. Mersey [39] found that the internet was not as powerful a builder of sense of community as print newspapers, recording a weakening of community institutions in the wake of the loss of a sole community newspaper in rural Arizona. Digital media also tends to be more partisan [40–42] due to consumers having more control over what they read in this relatively less curated space [35]. This is a factor that became of elevated concern during the pandemic, particularly in relation to public health information dissemination [43].

This reduction in trust is at least partially justified. Staff reductions, an indicator of a shaky community newspaper, see stressed journalists increasingly rely on sources other than their own investigation, notably agency work [44] and public relations/press releases [45–47]. There is also a global tendency for media companies to form strategic alliances to improve efficiency [48]. Even in markets experiencing significant disruption, there has been a concentration of media sources of information. For example, in the small and relatively diverse Swiss newspaper market, a textual analysis shows that in recent years, there has been a convergence in content, particularly in the coverage of international news [49]. Scholars agree on the value of diversity in news content [50] and ownership [51] in maintaining political dialogue and improving the quality of policy outcomes.

Finally, it is worth examining which sector of the population the loss of print impacts the most. American research suggests that those regions that have been left with no papers are the poorest, least educated, and most geographically isolated [52]. Newspaper reader demographics are collected and published diligently by publishers keen to attract advertisers drawn to specific demographics, and thus these data vary in integrity.

A better understanding of reader characteristics can be found in government statistical data, and here, good-quality data are particularly scarce for the developing world. However,

evidence from national American studies, for example, shows the median age of print newspaper readers is 53.5, while those viewing news on mobile phones is 38.6 [53]. A recent UK analysis shows that in the over 65 category, print newspapers remain a key means to access news (with 60% naming it as a main platform), while for those aged 16–24, by contrast, the internet dominates (82%) [54]. Available Australian data indicate a similar pattern: older Australian Bureau of Statistics data showed that those reading print newspapers at least once a week fell proportionately with age [55].

## 3. Case Study: The Pre-and Post-Pandemic Australian Print News Landscape

### 3.1. Historical Backdrop

Australia has a short print history relative to Europe and the United States, but with a surprisingly vibrant rural and regional media dimension, considering the low population density of non-metropolitan regions. Relative to the metropolitan dailies that dominate the newspaper landscape today, the Australian newspaper market developed a regional and rural presence surprisingly rapidly. Kirkpatrick [56] notes that only in Western Australia was there a substantial delay between the first publication in the state capital, and the first publication of newspapers in country towns (38 years). In other states (which were then independent colonies), the gap between the emergence of a city and a country press was less than 15 years. Only 400 people lived in the Corio Bay settlement of Geelong in 1840 when the *Geelong Advertiser* (which still exists) began publication—just two years after a paper was launched in Melbourne. Newspapers were established at a time in Australia when the idea of a free press was by no means settled or accepted by the government [57]. The first provincial newspaper appeared in the island state of Tasmania in 1825. Lack of profitability rather than government interference appeared to lead to a very high turnover of regional and rural titles during the early years.

### 3.2. The 21st Century and the Pandemic: Gradual and Sudden Losses

After the early rapid increase in titles in both metropolitan and rural/regional markets, the change in the population of titles settled, profits were made, and further major challenges to the power of print only arose with digitalization later in the 20th century. Australian newspapers took the same path as global operators, seeking to monetize digital offerings by placing paywalls on mastheads or key stories, with the consequent revenue seemingly unable to sustain the same level of journalism staffing [58].

As a result, most Australian cities have gone from at least two daily mastheads to one; now only the two largest cities in Australia, Melbourne and Sydney, have more than one daily. In the US, the number fell from a peak of almost 700 cities to under a dozen by 2010 [9]. Similarly in Australia, 121 dailies were published in 47 rural and regional centres at the peak before the decline in the 20th century. This loss of pluralism has come with an absolute loss of diversity of voices, even in the urban market. The newsrooms of major urban dailies have shrunk significantly. The two largest players, News Corp and Fairfax, cut 3000 jobs between them in 2012 alone [59], proportionately fewer jobs than in the US or Britain during the same period. For example, from 2008 to 2017, the decline in newspaper newsroom employees in the US fell by 45% [60].

In Australia, the job losses of the first two decades of the 21st century were minor, when the single year of 2020 is used as a comparator. The Australian Newsroom Mapping Project indicated that in 2020, 194 newsrooms showed a net contraction in staff, with 66 enjoying some expansion [61]; this number includes non-print operations. While the job losses were hard to quantify exactly due to limited transparency of commercial information, job losses in the arts and media totalled 22,200 nationally in the first four months of 2020, second only to job losses in the hospitality sector [62]. There were temporary halts in printing in many smaller rural and regional sites, with News Corp and ACM/Fairfax leading the way [62]. In many of the larger towns and regional cities, News Corp maintained a digital presence, but these titles maintained no office, and the news was largely generated by journalists geographically located outside the region on which they were reporting.

The Australian Communications and Media Authority states in its opening statement "we encourage diversity in Australian broadcasting services", but its encouragement is exercised at arm's length through the Broadcasting Services Act 1992 which sets limits on the control of commercial TV, radio, and newspapers. In fact, despite the encouragement of the ACMA, Australia's newspaper industry has in recent years been ranked the world's third least diverse, behind two nations with media significantly in state ownership or control, namely China and Egypt [63]. Australian media ownership, however, was very much in private hands and dominated by three families over generations: the Murdoch, Packers, and Fairfax families [64]. A high newspaper proportion of this ownership concentration, particularly in the last two decades, can be explained by Rupert Murdoch's News Corp alone, which, at its peak, held 57% of the newspaper market by circulation [65].

There have been no global comparisons since 2016 looking at diversity, but the Australian market was fundamentally changed in 2020, ostensibly due to the pandemic. However, ironically, the fall in circulations has meant a decrease in the dominance of News Corp, according to one analysis [65]. The large news networks in the five years prior to the pandemic had changed hands and ownership structure in a rapid sequence of transactions, and it was these new reformatted networks that experienced the large proportion of both job and title losses in 2020. The sudden rush of newspaper ownership changes and mergers with the three remaining big players in the nation, Australian Community Media (ACM), News Corp Australia, and Nine, agreeing to print each other's newspapers to save costs [66]. In the following section we will look more closely at what the data suggest how this lopsided collapse impacted differently in rural and regional Australia.

*3.3. Data Trends*

The emerging big player in regional newspapers, the Star News Group, snapped up rural titles that had been historically owned by the Fairfax group in late 2022 [67]. Apart from a further reshuffle in ownership, new newspaper titles also emerged. A Google Ngram frequency search for the phrase "new newspaper" in the corpus of books and magazines included in the Google database shows a sequence of peaks and troughs in the frequency of the term, with the peaks being separated by troughs of increasing size. The last major peak in the appearance of the term was in 1955, which coincides with the date of the launch of the last new daily newspaper in regional Australia, a record ironically only broken post-pandemic.

In the latest data, the term has declined to the lowest level seen in over 110 years. However, Google Ngram does not account for the extraordinary events of 2020. If one limits one's search to just the last 12 months, and limits it to the single publishing context, Australia, the search for the term "new newspaper" produces over 50 results, barring results excluded for repetition. Set the search for the previous 12 months, and the results return just 5. Curiously, the pattern of 6 or fewer results per year was broken only once before the pandemic, in 2019, with 13 results, before the numbers quadrupled again in 2020. Exploring further the contents of these 2020 results shows that almost all of these titles appeared in rural and regional locations [68]. This is part of a global pattern: hyperlocal and in particular rural media appears to have been partly insulated against the collapse in advertising spending over the last two decades [6]. One of the correlates of rurality, for many in the readership catchment of these titles, is inferior internet access, meaning digital options (for example in rural Australia) are not currently viable. Rural Australia is also demographically different to metropolitan Australia, in particular, older, thus with a greater history of print familiarity and lower levels of digital literacy.

A second, more direct approach to examining where the revival appears to have happened is The Australian Newsroom Mapping Project database [54], which provides a more comprehensive but still not exhaustive overview of the revolution that has occurred. Dickson described it as an 'extinction event', but it was a crisis disproportionately experienced by the big three, most notably News Corp. An analysis of the Australian Newsroom Mapping Project database [61] shows that of 70 of the 287 titles that experienced growth

in newsroom resources during 2020, 19 were News Corp products, while of the 217 titles that experienced contraction, 119 were owned by News Corp—almost exactly double the proportion. Of the 67 new mastheads that appeared in 2020, 45 had ownership structures outside the major newspaper chains, a pattern which we will return to in the next section.

The pandemic thus appears to have led to a closure of newspaper titles completely unprecedented in Australian history, with our analysis showing a greater impact on larger media conglomerates.

However, if the change in media titles has hit the regions harder, then, as noted earlier, this change would have a great impact, as these regions are relatively more reliant, due to demographic and technology access reasons, on print. A study into the state of regional newspapers was undertaken by the Australian federal government during the pandemic. Dr. Anne Webster, chair of the inquiry, explained the role of regional newspapers in the sustainability of their communities, calling them a "shared community experience [69]. "Regional newspapers", she noted, "put into the public area the stories of people who have excelled in sport, school or their business enterprise, as well as coverage of valuable community-based issues such as council decisions, court matters, public health issues and local weather events" (p. v). During the pandemic, reliance on online rather than print sources of papers may have contributed to a profusion of conspiracy theories, with particularly low COVID-19 vaccination rates experienced in rural and regional centres, relative to the metropolitan areas.

As an ACCC analysis shows, 92% of metropolitan local government areas were serviced by at least one newspaper at some point during the period 2008–2018, compared to 59% of regional areas. Our analysis of 2023 data in the Public Interest Journalism Initiative database shows that the number of 'news deserts' has, if anything, shrunk [54]. It is worth noting that the pandemic is unique in terms of its impact in Australia. During the First World War, the last occasion of a significant global pandemic, as well as economic and social crises associated with both the war and public health, there was no such fall in the number of newspapers circulated [56].While this suggests that the 2020 pandemic appears to have struck the industry at a point of vulnerability, the disproportionate loss of titles belonging to the fiscally more stable major players suggests the decision to close print sites may have been more strategic than imperative. The executive chairman of News Corp made the closure announcement by blaming COVID-19, which he said "has impacted the sustainability of community and regional publishing" [70]. However, he noted the company planned to redesign its business into a more digital shape. The closure of papers was accompanied by the closure of presses, with some News Corp print sites going from 24 houraday operation to shut down, overnight The 'sustainability' that News Corp was talking about, however, was financial sustainability, with the company experiencing a US$1.26 billion loss in 2020, turning that around to a $US330 million profit in 2021.

The Webster report [69] concludes that the closure of local newspapers has left residents "feeling mistrust and unattached to their communities" (p. 3). The Local News Consumers report indicates that regional areas bear the brunt of news deserts. Two-thirds of residents of local government areas with populations of less than 30,000 reported a decrease in information about their own communities, and one-third reported having "fewer topics to share with friends" [14], suggesting a significant decline in social capital and cohesion.

*3.4. What Do These 'New' Newspapers Look Like?*

With the pandemic stripping newspapers disproportionately out of rural and regional areas, and disproportionately impacting the large newspaper brands, what are the implications for the Australian newspaper landscape and its role in delivering on SDGs?

The Australian government has become a key player in regional media due to the creation of the Regional and Small Publishers Innovation Package, an AUD 60.4 million 3-year programme. The Package was released with a statement that indicates that the federal government has concerns about the regional newspaper industry. "The media

industry is in a significant and sustained transition period, putting the delivery of quality journalism under pressure. This poses challenges for small publishers and small regional newspapers in particular", the announcement of the package stated [63]. The stimulus package appears to have at least partially succeeded, and succeeded at adding new publishers to the Australian media landscape.

For instance, the *Lockyer and Somerset Independent* was launched as a print-only news source in Queensland in October 2020; the *People's News* was launched January 2021 in Mackay, Queensland; the *Nyngan Weekly* started in October 2020 published by Gilgandra Newspapers; the *Murray Bridge News* was launched by journalist Peri Strathearn after his employer, Australian Community Media, temporarily closed the *Murray Valley Standard* (Victoria) in April 2020; the *Ararat Advocate* (Victoria) was launched after another Australian Community Media (ACM) closure (the *Ararat Advertiser*). Publisher Craig Wilson originally intended it to be a temporary publication but confirmed to the Public Interest Journalism Initiative that it "would continue permanently" [54]. In addition, the *Daily Journal* appeared in Warwick as the first new Australian print daily in almost half a century, and, although it ceased daily production after six months, it continues as a twice-weekly two years later (The *Daily Journal* was launched by the authors in late 2020, but is now majority-owned by local parties).

However, the intent of these new media outlets appears to be modest, if one reviews the public statements of the new newspaper proprietors. The following brief analysis is based on public statements made by new newspaper publishers online during 2020, as well as non-exhaustive sample of statements made on radio and television during the same period.

The new newspapers are firmly 'community' in orientation, which we would define as working with the geographic community rather than serving a grander purpose 'for' the community. There is little evidence of a focus on 'hard' news; instead, a tendency to share positive community information and acting, as one new publisher called it, as a "communication engine" [71]. "There could be a profile on the new nurse, the owner of the pub, or the backpacker working at the store. I am pretty much working on keeping it focused on positive news and people" [72], one new publisher in a very remote region commented. Almost all of the public statements, in light of the mass collapse of print newspaper titles, mention the minimal benchmark of survival of 'a' paper in their region, rather than aspirations to tackle the issues that might otherwise be forgotten. "It's too early to be closing newspapers", said one publisher. "In fact some may never close if they are set up right, for the community they serve" [73]. As yet, there is little evidence in the public discourse of an understanding of an activist role for newspapers, or an understanding of their role as educators, standard-bearers or record-keepers. Instead, the focus on sheer survival is reminiscent of the struggle of the first wave of newspapers in Australia, which Kirkpatrick notes also struggled more with profitability than political cross-currents [56].

The struggle for survival of these 'green shoots' regional and rural newspaper titles is exemplified by the outlier new publication, the *Daily Journal*. It appeared in the market 9 months after the closure of the *Daily News* in the same footprint, and was partly initiated as a result of local pressure for the restoration of a daily paper, which the region had enjoyed for a century. However, the 9-month delay had seen the atrophy of essential local infrastructure such as a delivery crew capable of making morning newspaper rounds to local lawns, as well as the creation of new digital habits that shrank the daily circulation of the new product well below the last circulation numbers of its predecessor.

The Public Interest Journalism Initiative adds quantitative evidence to what is happening with these new newspapers. The overall newsroom staffing of Australian newspapers from January 2019 to January 2023 shows 294 newsroom contractions and 165 newsroom expansions. The bulk of these expansions (including new firms) are in rural and regional areas, and the content that is available is in the public domain, suggesting that the new titles are focused on survival rather than a news mission. Overall, the number of dedicated journalists in these newsrooms has decreased, and instead an increasingly large number of

enthusiast owner-operators with no newspaper heritage are taking on the role, again an echo of the first wave of rural/regional newspapers in Australia.

## 4. Conclusions

Newspapers predate television, radio, and the internet, but still hold surprising power in the current complex media landscape. In regional and remote areas, where television, radio, and internet reception are variable, newspapers hold surprising sway, and studies show the importance of print media in maintaining a community conversation, building social capital, and improving governance. Studies show that when a newspaper is lost, there are impacts on everything from engagement with politics to access to public health information, with particular demographics, such as the elderly, impacted differently.

The focus of this paper has been on Australia and other key English-speaking markets. The pandemic represents an exceptional moment in the history of media, particularly in Australia, with the pandemic coinciding with the collapse of print newspapers on an unprecedented scale. We present data showing that the collapse was concentrated on both the monolithic media chains and the smaller (regional and rural) markets. As the pandemic passes, it will be possible to separate the causes of the collapse: what can be allocated to pandemic-related triggers for closure (lockdowns and economic crisis) and what can be allocated to pre-existing structural weakness of print media. Clearly, the power of print was receding prior to the pandemic.

The vacuum that appeared in rural and regional Australia created a natural experiment replicating to a degree what existed in Australia two centuries earlier, but this time with the additional complexity of television, radio, and digital news. Data from the Australian Newsroom Mapping Project database [61] and our own analysis of public announcements show that dozens of community-driven print newspaper projects have sprung up to replace the 'missing' newspapers, despite the economic downturn associated with the pandemic.

The data are sparse but indicate that while diversity has clearly increased in the pandemic, and new independent players have rapidly emerged in the markets vacated by large network media, there are reasons to temper optimism. Firstly, in the pause between the closure of legacy media and the arrival of the new print options, habits have changed. The power of social media during the pandemic rose, and with it, its capacity to create 'imagined communities' [74] that crossed geographical boundaries: the authors observed the largest political rally in half a century in their rural home town, with the speakers using tropes and language imported, sometimes without any attempt at cultural translation, from American conspiracy theories on the pandemic and the loss of freedom.

The papers that may have reclaimed some of the 'newspaper deserts' the pandemic created are emerging in an environment of high levels of active government support, which reduces the true viability and independence of new players. Indeed, the presence of government support is an indicator of the weakness of the sector. When newspapers first developed, they attracted activist entrepreneurs interested in taking on issues of interest to the publishers and the public [56] with profit a happy side effect.

In the new media landscape, profit is not assured. The new independent operators are understandably focused on goals typical of new entrepreneurs, rather than the bolder 'mission' sometimes found in new news-oriented digital players. In-depth interviews with journalists in Europe and Australia about their motivation to engage in new digital media projects tends to show them driven by the activist, interpretative, and sense-making aspects of the journalist's role [75]. By contrast, the sparse data currently available on the new Australian community media appear to be driven by a relatively basic community information-sharing motive, meeting the need for a community noticeboard that Matthews [13] found was one of the key functions of a small regional or rural newspaper. This limited mission may evolve with time, but as the digital divide between rural and metropolitan regions reduces (for example, due to the increased availability of fast satellite internet) [76], it gives this new wave of print publishers little time to settle in their markets.

Despite these initial modest missions, the presence of this new cohort of print newspapers may ultimately still change their host communities more than their founders intended: community cohesion and public discourse provide a check on government whether or not the publishers categorise themselves as political activists. Newspapers have traditionally been regarded both in popular and scholarly analysis as both community-building and cathartic in that their role is to underscore the activities of locals as well as providing a conduit for revelations of corruption and political abuse [77]. The mere presence of print media may act to repress misbehaviour and improve governance [33].

**Author Contributions:** Conceptualization, O.M. and E.V.M.; Data curation, O.M.; Writing—original draft, O.M.; Writing—review & editing, E.V.M. All authors have read and agreed to the published version of the manuscript.

**Funding:** This research received no external funding.

**Data Availability Statement:** The data used in this study include publicly available data from datasets listed in the reference section. Some additional primary data was collected by the authors and is available from the corresponding author.

**Conflicts of Interest:** The authors declare no conflict of interest. As noted in the text, the authors formerly were majority owners of the Daily Journal newspaper through the Small Newspaper Company, but they no longer are controlling or majority owners.

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
