# Peer review of "Out of Print: What the Pandemic-Era Newspaper Crisis in Australia Teaches Us about the Role of Rural and Regional Newspapers in Creating Sustainable Communities"

_sustainability, doi:10.3390/su15065439_

Round 1
Reviewer 1 Report
an interesting theme about a newspaper with a sustainable community in the countryside. however, the analysis is rather in-depth and the argumentation is only based on the quantity of media before and during the pandemic in rural areas. It is better if the amount of content related to the issue of data sustainability is displayed. secondly, in the background of raising many background issues in America, is there any relationship between the mass media in America and Australia towards this issue?
Author Response
Reviewer 1’s Comments:
An interesting theme about a newspaper with a sustainable community in the countryside. however, the analysis is rather in-depth and the argumentation is only based on the quantity of media before and during the pandemic in rural areas. It is better if the amount of content related to the issue of data sustainability is displayed. secondly, in the background of raising many background issues in America, is there any relationship between the mass media in America and Australia towards this issue?
Response to Reviewer 1.
Thank you for your feedback. You are advancing a number of useful points here, but firstly to address the question that closes you feedback: While there are indeed similarities between the decline in newspaper readership in the United States and in other western countries, there are significant differences between the mass media in the American market and that of Australia which is possibly outside the scope of this study (and the authors, as a former American and former Australian journalist are in a reasonably good position to describe these differences.
Perhaps most strikingly however, ownership concentration in the Australian market is the greatest point of differentiation between the US and Australian markets. Australia’s market was traditionally divided into three major players, but now the majority of the newspaper market lies in two hands. As this paper notes, however, amongst the smaller operators, there is greater diversity. In addition to this difference in ownership concentration, the most significant barrier to entry in the Australian market are the geographic challenges both of distribution and of obtaining newsprint (due to relatively lower volumes there is only a single domestic supplier of newsprint in Australia). We have, however, added evidence showing that the loss of social capital found in American studies has been mirrored in Australia.
However, turning to the other key point, the reviewer notes that the analysis as it stood was focused more on quantity before and after the pandemic. We have addressed this point to push the balance in the direction of discussing qualitative changes in the market. Section 3.3 and 3.4 Data Trends has been altered and expanded. The reviewer will also note significant additional changes have been made to the manuscript, adding data/evidence to the case study analysis section.
Reviewer 2 Report
This article studied the degree of print newspapers part of the solution to sustainable rural and regional communities, using a case study approach. However, there are the following specific shortcomings.
(1) First of all, the abstract is too simple, and does not summarize the article's conclusions or explain what the value of the article's research is. After reading it, I do not know what the innovation of the article is.
(2) Secondly, please check the basic format of the article carefully before submitting it! There are a lot of blank spaces in the article (as shown in the legend below) and the legend lacks punctuation. Besides, punctuation is missing at the end of section 2.3. Please adjust the article format appropriately.
![]()
(3) Finally, the subsections of the third part are not logical enough and it is difficult to understand the connection between each part. Please add appropriate logical paragraphs for clarification.
(4) The phrase "developing countries" at the beginning of the last paragraph in section 2.4 contradicts the examples of countries that follow (America, UK, Australia, etc.)
In my opinion, This article mainly has many flaws in details. Major review before publication.
Author Response
Reviewer 2’s comments:
(1 First of all, the abstract is too simple, and does not summarize the article's conclusions or explain what the value of the article's research is. After reading it, I do not know what the innovation of the article is.
(2) Secondly, please check the basic format of the article carefully before submitting it! There are a lot of blank spaces in the article (as shown in the legend below) and the legend lacks punctuation. Besides, punctuation is missing at the end of section 2.3. Please adjust the article format appropriately.
(3) Finally, the subsections of the third part are not logical enough and it is difficult to understand the connection between each part. Please add appropriate logical paragraphs for clarification.
(4) The phrase "developing countries" at the beginning of the last paragraph in section 2.4 contradicts the examples of countries that follow (America, UK, Australia, etc.)
Response to Reviewer 2's comments:
Thank you for your feedback. We acknowledge the abstract while summarizing the scope of the project, completely overlooked the conclusions, and we have rectified this—thank you again. In relation to formatting, this indeed was an irritating characteristic of the manuscript—we are unsure why this occurred and why it was overlooked, and we have addressed this in the revised manuscript. The manuscript as you will note has been significantly revised and we hope that overall the underlying thesis of the manuscript is clearer: that print news has played a unique and important role in supporting the sustainability of communities, and in delivering outcomes ranging from the building of social capital to improve political, economic and health outcomes, and that the pandemic has left the sector seriously weakened.
Re the use of ‘developing countries’, thank you for pointing out that error. We have re-written that sentence to indicate that data is scarce in developing countries however there are statistics deriving from American, UK and Australian sources. We have also noted this limitation in the conclusion.
We have tweaked the language and linkages between the subsections in Section 3 to hopefully make the thread of the manuscript’s narrative clearer. Overall, the manuscript has been significantly edited and additional evidence added to strengthen the case study.
Reviewer 3 Report
This paper discusses the decline/closing of print newspapers in developed countries with special focus on Australia in the context of Covid-19 and its relationship with sustainable rural and regional communities. The authors argue that the closing of print newspapers in the region affect the conversations between certain communities, the visibility of those communities in the national as well as international level, social capital, local and national governance. It is a significant intervention in the studies on print and sustainable communities. While most of the recent observations concentrate on the problems of printing due to the environmental and sustainability questions, this paper explores the questions of remote communities and the need for print journalism. However, the paper requires some revisions to highlight this point as it is not elaborated with examples.
For instance, though the paper is about the Australian case study, less space is given for that discussion. Though the extensive literature review is helpful to contextualize the study, significance should be given to primary materials and their discussion. One of the ways to do this is to give individual examples of newspapers from Australia and how they are helping in sustainable community development. Similarly, though the paper mentions the elderly population in the beginning, there is hardly any discussion on how they are coping with the decline of print newspapers and the digital boom. Moreover, it would be better to focus on the socially marginalized communities and their engagement with print newspapers in the region. The question on new players should also be elaborated with examples. Some of the references are missing in the paper, especially to support the claims made by the authors. The third section (Australian case study) should be reworked with examples.
25-27 “…but there is a substantial body of work delineating the relationship between the presence and use of print newspapers in developing community literacy and social capital, and the role that literacy and social capital play in turn, in delivering SDGs.” (Please add references)
80-82. “Scholarly and non-scholarly debate on the importance of local print newspapers indicates community sustainability is threatened by the loss of regional titles.” (References)
33 “In almost every market, bar India, circulations continued to shrink” It would be better to suggest in passing why India is an exception.
111-113. “In Life is Harder, the regional editor of a series of daily newspapers describes a recent, extended study of one community in Bowling Green, Virginia, which suddenly and without notice, lost a 99-year-old newspaper in 2019.” Which community and which newspaper?
"Digital media tends to be more partisan" – This could be said about print journalism as well. Benedict Anderson’s Imagined Community explains this question. Instead of dismissing digital media, one could ask why digital media creates distrust among people. One of the reasons would be the abundance of information/misinformation which was not the case during the print days. The ephemerality of news in the contemporary times is also another factor. It would be better to give a little more historical details to stress on the significance of print communities. For instance, how coffee clubs and tea stalls worked as public reading and discussion spaces. Then the argument on digital media and individual culture would be stronger.
87-91 “Finally, this analysis shows how, remarkably, the pandemic appears to have revitalized, at least temporarily, independent press in a market that has been traditionally one of the least diverse in the world and offers caveats on optimism that might arise from this resurgence in relation to the delivery on the SDGs.” This point is not elaborated well in the article. It would be good if the authors could give specific examples of these independent presses than giving only a generalized picture.
366-369. “our own analysis of public announcements that dozens of community-driven print newspaper projects sprung up to replace the ‘missing’ newspapers—despite the economic downturn associated with the pandemic. (Examples?)
317-318 "the federal government is has concerns about the regional newspaper industry." (Language)
Author Response
Reviewer 3’s comments
This paper discusses the decline/closing of print newspapers in developed countries with special focus on Australia in the context of Covid-19 and its relationship with sustainable rural and regional communities. The authors argue that the closing of print newspapers in the region affect the conversations between certain communities, the visibility of those communities in the national as well as international level, social capital, local and national governance. It is a significant intervention in the studies on print and sustainable communities. While most of the recent observations concentrate on the problems of printing due to the environmental and sustainability questions, this paper explores the questions of remote communities and the need for print journalism. However, the paper requires some revisions to highlight this point as it is not elaborated with examples.
For instance, though the paper is about the Australian case study, less space is given for that discussion. Though the extensive literature review is helpful to contextualize the study, significance should be given to primary materials and their discussion. One of the ways to do this is to give individual examples of newspapers from Australia and how they are helping in sustainable community development. Similarly, though the paper mentions the elderly population in the beginning, there is hardly any discussion on how they are coping with the decline of print newspapers and the digital boom. Moreover, it would be better to focus on the socially marginalized communities and their engagement with print newspapers in the region. The question on new players should also be elaborated with examples. Some of the references are missing in the paper, especially to support the claims made by the authors. The third section (Australian case study) should be reworked with examples.
"Digital media tends to be more partisan" – This could be said about print journalism as well. Benedict Anderson’s Imagined Community explains this question. Instead of dismissing digital media, one could ask why digital media creates distrust among people. One of the reasons would be the abundance of information/misinformation which was not the case during the print days. The ephemerality of news in the contemporary times is also another factor. It would be better to give a little more historical details to stress on the significance of print communities. For instance, how coffee clubs and tea stalls worked as public reading and discussion spaces. Then the argument on digital media and individual culture would be stronger.
25-27 “…but there is a substantial body of work delineating the relationship between the presence and use of print newspapers in developing community literacy and social capital, and the role that literacy and social capital play in turn, in delivering SDGs.” (Please add references)
80-82. “Scholarly and non-scholarly debate on the importance of local print newspapers indicates community sustainability is threatened by the loss of regional titles.” (References)
33 “In almost every market, bar India, circulations continued to shrink” It would be better to suggest in passing why India is an exception.
111-113. “In Life is Harder, the regional editor of a series of daily newspapers describes a recent, extended study of one community in Bowling Green, Virginia, which suddenly and without notice, lost a 99-year-old newspaper in 2019.” Which community and which newspaper?
87-91 “Finally, this analysis shows how, remarkably, the pandemic appears to have revitalized, at least temporarily, independent press in a market that has been traditionally one of the least diverse in the world and offers caveats on optimism that might arise from this resurgence in relation to the delivery on the SDGs.” This point is not elaborated well in the article. It would be good if the authors could give specific examples of these independent presses than giving only a generalized picture.
317-318 "the federal government is has concerns about the regional newspaper industry." (Language)
366-369. “our own analysis of public announcements that dozens of community-driven print newspaper projects sprung up to replace the ‘missing’ newspapers—despite the economic downturn associated with the pandemic. (Examples?)
Response to Reviewer 3’s feedback.
Thank you for your thorough feedback, and in particular, picking up errors and omissions. We acknowledge that the original manuscript did not entirely deliver on its promise. We have extensively edited the manuscript, adding additional evidence and cases, and also working on the narrative in the early sections to ensure the ‘message’ of the case study is clearer.
We have, in response to your suggestions, added a significant amount of detail as well as additional references specifically on the Australian case, including the use of individual examples as you have suggested. We have picked up on the issue of the elderly and their digital literacy to ensure this (key) point is not lost later in the manuscript. Thank you for picking this up.
The one area where we didn’t follow your advice (apologies!) as in your call for references in the region of lines 25-27, as these references do follow in Section 2 and are extensive—indeed to cumbersome to place here, we feel. We have however addressed the issues of grammatical errors and the lack of name of the US paper in the case study quoted (The Caroline Progress). We have added examples where examples were called for and added some additional references particularly focused on the Australian case.
The comment on digital media and partisanship is particularly interesting and welcome, and Anderson’s Imagined Communities, in a sense pre-digital, is a very interesting lens with which to observe the transformation of communities during this era of media change. Digital has in fact also ‘expanded’ and even ruptured sense of community, with American terminology and conspiracy theories easily slipping into rural Australia during the pandemic. In our home town of Warwick, the largest political meeting of the last half century, featuring a line-up of conspiracy theorists, took place in the midst of COVID, with rhetoric drawn from an American context dropped almost unchanged into a rural Australian context. Even though this example was simply observed as journalists, we have added this to the conclusion. In addition, we have altered the manuscript to try to reflect (where evidence is available) this, if you can call it that, geographical anachronism facilitated by digital (largely social) media.
Thank you again for your feedback.
Round 2
Reviewer 2 Report
The authors addressed all the comments. I accept the paper in its present form.